# A Cluster-Randomised Crossover Pilot Feasibility Study of a Multicomponent Intervention to Reduce Occupational Sedentary Behaviour in Professional Male Employees

**DOI:** 10.3390/ijerph18179292

**Published:** 2021-09-02

**Authors:** Gail Helena Nicolson, Catherine B. Hayes, Catherine D. Darker

**Affiliations:** Public Health & Primary Care, Institute of Population Health, Trinity College Dublin, Russell Centre, Tallaght Cross, D24 DH74 Dublin, Ireland; hayesc9@tcd.ie (C.B.H.); catherine.darker@tcd.ie (C.D.D.)

**Keywords:** sedentary behaviour, multicomponent intervention, workplace, socio-ecological model, males

## Abstract

Professional male office employees have been identified as those most at risk of prolonged sedentary time, which is associated with many long-term adverse health conditions. The aim of the study was to assess the acceptability and feasibility of a gender-sensitive multicomponent intervention, guided by the socio-ecological model, to reduce occupational sedentary behaviour by increasing physical activity in professional men. The main elements of the intervention comprised: a Garmin watch with associated web-based platform/smartphone application, an under-desk pedal machine, and management participation and support. A cluster-randomised crossover pilot feasibility trial recruiting professional males was conducted in two workplaces. Mixed methods were used to assess the primary outcomes of recruitment, retention, and acceptability and feasibility of the intervention. Secondary outcomes included objectively measured sedentary behaviour, standing and physical activity. Focus groups were used to explore the acceptability of the intervention in a real-world setting. Twenty-two participants were recruited (mean age 42.9 years (SD 11.0)). Recruitment and retention rates were 73.3% and 95%, respectively. Overall, participants found the intervention acceptable and feasible, and expressed enjoyment of the intervention, however desk set-up issues with the pedal devices were noted. The manual recording of the pedalling bouts was overly burdensome. Preliminary data indicate that the intervention may reduce occupational sedentary behaviour and increase physical activity. This intervention should be further tested in a definitive trial following consideration of the findings of this pilot feasibility trial.

## 1. Introduction

Sedentary behaviour (SB) is strongly associated with adverse health outcomes, and prospective studies have indicated that longer time spent being sedentary is associated with all-cause mortality, cardiovascular disease mortality, and type 2 diabetes [1]. Occupational SB has been found to be associated with cancer incidence and mortality [2,3]. Urban-located professional male employees have been identified as those with the longest sitting times (>7.5 h per day), and the workplace setting is the context in which SB is mostly accrued [4]. The office workplace offers several advantages as a setting for interventions to reduce daily SB due to the opportunity to reach a large working population, and where multiple influences that promote SB can be targeted [5].

The socio-ecological approach to designing health promotion interventions advocates targeting all important influences on behaviours, e.g., individual, interpersonal, environmental, and organisational determinants of workplace SB [6]. Advances in digital tools such as smartphones, internet-based platforms and consumer wearable technology are useful methods to support and target individual behaviour change techniques (BCTs). Interventions to reduce SB which have adopted goal-setting, self-monitoring, education, and feedback have been deemed as the most promising [7]. Having a goal can serve a directive and energising function and positively affects persistence and action [8]. A powerful technique to disrupt habits is to bring the habitual behaviour and its context into conscious awareness [9]. This might be achieved by means of self-monitoring [10], and mobile technology has been noted as an effective method to incorporate self-monitoring in behaviour change interventions [11]. Michie et al. [12] define feedback as the reinforcement of performance of the specific targeted behaviour. Irrespective of the target behaviour or technology used in intervention studies, different types of feedback have been found to be effective to change habits [10]. For example, van Dantzig et al. [13] provided descriptive and persuasive feedback to participants’ smartphones whenever 30 min of uninterrupted computer activity (used as a proxy for SB) was recorded, to break their SB.

Environmental restructuring [7], and the use of digital prompts to encourage breaks in sitting has produced promising results [14,15,16,17]. Prolonged SB triggers a state of metabolic ‘inflexibility’, even among individuals who meet PA recommendations, by disrupting fuel homeostasis and metabolic health [18]. Frequent interruptions to SB with bouts of activity (even 1 min duration) have been associated with improved metabolic outcomes, including in those who exercise regularly [19,20]. Thus, breaking up time in SB is a stimulus for improving metabolic health (flexibility) and has been suggested as a novel and promising strategy in the general population [21]. This is especially relevant in settings where SB is widespread such as office workplaces and may help reduce the risk of and prevent chronic diseases. Many workplace interventions centring on SB have examined the use of sit-stand workstations [22,23]. Although the act of standing up expends some energy, very low levels of energy (≤2 metabolic equivalents (METs)) are expended during quiet standing [24]. No changes in BMI, body mass, body fat and lean mass [25], or reductions in postprandial glycaemia have been found as a result of standing [26], compared with engagement in low intensity physical activity [27]. Furthermore, standing for long periods may invoke deleterious outcomes for cardiovascular health [28], and has been found to be associated with an increased risk of ischemic heart disease and varicose veins [29,30]. It may be argued that if standing is the primary objective of SB interventions, minimum, if any, health benefits from a public health perspective may be observed at this level of energy expenditure [31]. Greater intensity activity may be required to invoke meaningful health benefits [32,33]. Pedalling an under-desk device has the benefit of allowing employees to continue computer-based work tasks [34], important in terms of productivity, and even very light to light effort while pedalling (30–50 watts) a stationary bike has been found to expend 3.5 METs [35]. Recent reviews of the literature [36,37] investigating metabolic markers in the prevention of type-2 diabetes have called for further exploration of the effects of very light intensity breaks of short duration to address concerns about productivity, practicality issues, the habitual nature of workplace SB, and management support. Psychosocial support from colleagues and managers in workplace interventions may positively influence the motivation, participation and adherence through a norm-changing social supportive culture [38], and is a frequently observed BCT used in behaviour change interventions [7]. The importance of this strategy has been highlighted in previous SB intervention studies [39], and a focus on management support as well as organisational-level change is fundamental as part of a ‘whole-systems approach’ [40]. 

Although the root causes of occupational SB are similar for both genders—i.e., restrictive workstations and the traditional workplace culture of remaining seated for long periods—in terms of intervention participation, men are especially difficult to recruit to health promotion interventions [41]. Bottorf et al. [41] describe males as a ‘hard-to-reach’ population where specific challenges lie in implementing illness prevention and health promotion initiatives such as physical activity (and SB). Gender responsiveness in the design of interventions to prevent non-communicable disease is advocated by the World Health Organisation [42]. In interventions to increase PA, the most effective outcomes have been observed in gender-sensitised interventions that recognise men’s interests and tailor health promotion efforts for this group [41,43]. For example, a holistic approach that includes healthy diet and relaxation or wellness may be preferable to women [44], whereas men may favour competitive and exercise oriented activities [44], and interventions that require low time commitment [41]. In terms of intervention design, at an interpersonal level, the facilitation of social comparison ‘involves explicitly drawing attention to others’ performance to elicit comparisons’ (pg.1493) [45], and has been found to be effective in PA interventions [41]. Social comparison using friendly competition and self-monitoring have been found to increase PA in men [41,43]. Compelling evidence exists that well-designed interventions for men can lead to positive behaviour changes [46]. 

In line with best practice recommendations from the Medical Research Council [47,48], a participatory approach to ensure context-appropriateness of components, and consideration of the end-user preferences was adopted in the development phase of this intervention. The provision of insights from the target population of professional males, who are under-represented in health promotion interventions, and the involvement of key stakeholders, i.e., employees, managers and managing partners, were deemed essential in the development and evaluation of this workplace intervention. Their voices provided a crucial understanding of the practicalities experienced by participants that is essential in developing ‘useable’ interventions for health. Combining the BCTs goal-setting, self-monitoring, social comparison and digital prompts in a multicomponent intervention using mHealth technology, providing education and weekly feedback, together with an under-desk pedal machine, and social support by recruiting managers to participate in the intervention, to reduce occupational SB in a male only sample, has not previously been investigated. Therefore, the primary aim of this pilot feasibility study was to refine the intervention content using mixed methods to assess the acceptability and feasibility of the intervention components and trial measures. This will optimise the format for real-world implementation and evaluation by identifying key methodological and implementation issues that need to be addressed prior to effectiveness assessment in a future definitive cluster randomised controlled trial.

## 2. Materials and Methods

### 2.1. Participants

Twenty-two office-based employees from two professional worksites in Dublin, Ireland were recruited. Recruitment involved a two-step process. Convenience sampling was employed to recruit professional organisations. Step 1: in total, managers in five organisations were approached through the researcher’s personal networks and invited to participate in the study. Two organisations agreed to participate, and permission was obtained to contact male employees to inform them about the study. Step 2: to recruit eligible participants, purposive sampling was used via emails sent by a contact within each company (one a manager; and one lead for corporate and social responsibility). Participants included members of management and managing partners, as well as employees. No remuneration was given to participants.

Inclusion criteria for Step 1 were professional urban-based organisations, and for Step 2, adult men who spend the majority of their working week performing seated desk-related activities. Exclusion criteria were:FemalesAged under 18 yearsThose with contraindications or limitations to physical activity as indicated by the Physical Activity Readiness Questionnaire [49]Those without a personal deskThose who planned to be absent from the workplace for more than two days in one week during the study periodThose who were involved in another sedentary behaviour reduction programme or intervention.

All participants were given a participant information leaflet and asked to sign a consent form.

### 2.2. Study Design and Procedures

Ethical approval was obtained from the Research Ethics Committee of the School of Medicine, [BLINDED FOR REVIEW] (ref. 20190702). The pilot feasibility study was conducted between October and December 2019. This study was a cluster-randomised crossover trial, consisting of two arms: Intervention and Control. The crossover comprised a two-week ‘Cycle at Work’ intervention period and a 2-week control period, separated by a one-week washout/usual habits period. Participants were randomly allocated to one of the two clusters on a 1:1 basis. Details of the protocol have been previously published [50]. All elements of the study were conducted on-site in participants’ place of work. A statistician, who was not involved with the study, determined simple cluster randomisation by using randomisation software to allocate each worksite to begin with the intervention or control period. Group allocation was concealed until after baseline assessments were completed. Due to the nature of the study, i.e., environmental restructuring, blinding of group assignments was not possible. Participants were fitted with an activity monitor to measure their baseline SB, standing and physical activity, which was worn continuously for nine consecutive days. Contextual and modality information on sedentary behaviour and physical activity were collected using ecological momentary assessment (EMA) (https://pielsurvey.org; v1.2.4.2; accessed 2 September 2019) downloaded to each participants’ own smartphone, and anticipated benefits of the intervention, and work engagement [51] were also collected at this time point using questionnaires (Figure 1). Following randomisation, a buffer week was required for logistical and practical reasons for the intervention set-up. Participants randomised to the intervention period were provided with a compact stationary under-desk pedalling device (DeskCycle2 model; 3DInnovations LLC, Greeley, CO, USA), and a Garmin Forerunner 35 PA tracker watch for the full intervention duration (2 weeks), whilst participants in the control trial did not receive the intervention equipment and were asked to maintain their normal workplace habits. To measure pedalling times, as there is currently a lack of commercially available devices that accurately detect under-desk cycling and provide the user with immediate feedback, a Bluetooth cadence sensor in conjunction with manual recording and subsequent uploading via the Garmin watch was necessary. The washout period was identical to the control period; however, no measurements were taken, and no contact was made with participants by the researcher. All measures were repeated in the control and intervention periods. Acceptability and feasibility of the intervention were evaluated directly after the study ended (8-weeks). This study was guided by the TIDieR checklist for intervention description [52] and structured using the updated CONSORT guidelines for reporting feasibility trials [53]. An adapted CONSORT flow diagram is presented (Figure 2). Figure 1 illustrates the participants’ flow in the overall ‘Cycle at Work’ study.

### 2.3. Intervention Description of ‘Cycle at Work’

The Cycle at Work intervention targeted multiple components. Initially, an education session was delivered to participants by the primary researcher on the dangers associated with prolonged SB and the potential benefits of reducing SB. To target environmental-level influences of workplace SB, participants were provided with an under-desk pedal device to enable light physical activity throughout the workday to interrupt SB. A Garmin Connect account was set up for each participant, and teams were allocated by the researcher within the platform prior to the study commencement. Permission to access participants’ account throughout the study was granted, and at the end of the study period participants were advised to change the passwords to the accounts. Setting SB goals was not possible on the Garmin Connect platform, therefore, cycling/pedalling time goals of 30–40 min per workday were set for each participant. Manual measurement of pedalling times using the Garmin watch facilitated self-monitoring to increase conscious awareness of breaking SB with LPA. After recording and uploading, pedalling times could be observed on the Connect platform allowing social comparison and friendly competition among the men. The principal researcher provided encouragement and feedback on participants’ activity progress via weekly emails. Segments appeared on the Garmin watch every 15 min of inactivity on its ‘move bar’, which accumulated to provide a sound and vibration alert after one hour of sedentariness and served as a digital prompt. Participants were required to engage in some physical activity (i.e., record stationary pedalling) to reset the move bar. Managers were recruited to participate in the intervention study. This was intended to provide employees with social support and facilitated a shared experience of reducing occupational SB in the intervention.

### 2.4. Primary Outcomes—Acceptability and Feasibility

Mixed methods were used to assess processes such as feasibility of recruitment, consent to randomisation, retention, randomisation procedures and to explore the feasibility, acceptability and participants’ experience of the intervention and study processes overall. Recruitment and retention logs, and information on eligibility were recorded for assessment of feasibility outcomes. Assessment of acceptability of the user experience of the intervention, and the study measures and processes overall were evaluated at follow-up using focus groups and a one-to-one semi-structured interview by GN (female, PhD student), who had experience in conducting focus groups and one-to-one interviews. A prior relationship had been established with the participants who were involved in the development process of the intervention. A semi-structured questioning schedule was used incorporating the following themes: individual intervention components such as the under-desk pedal device, the mHealth components, and acceptability of the overall intervention from management and employee perspectives. The interview guide was pilot tested in a convenience sample of research colleagues in the Discipline of Public Health & Primary Care^1^ and was adapted where necessary. Prompts were used to keep the flow of conversation going if this did not happen spontaneously. Only the researcher and participants were present during the focus groups/interview which lasted 30–40 min each. Field notes were taken during and after the focus groups and interview sessions. After each focus group session, participants were debriefed by the researcher. Focus groups and the semi-structured interview were audio-recorded and transcribed verbatim. Transcripts were not returned to the participants, and feedback on the findings was not provided. A pen and paper implementation questionnaire was used to measure acceptability, feasibility, and appropriateness as they are seen as the forerunners of indicators of implementation success [54].

### 2.5. Secondary Outcomes

#### 2.5.1. Sedentary Behaviour, Standing, Physical Activity

The secondary outcome measures included objective measurement of changes in SB and PA, and pedalling time at three time points, T0 (baseline), T1 (1-week post baseline) and T2 (5-weeks post baseline).
Total sedentary behaviour: waking hoursTotal sedentary behaviour: work hoursTotal physical activity: waking hoursTotal physical activity: work hoursPedalling time: work hours

Sedentary behaviour and physical activity were assessed at baseline (before randomisation) and throughout the control and intervention periods. Key recommendations when using the activPAL3 monitor in field-based research by Edwardson et al. [55] were used and the full description of the activPAL procedure is outlined in the study protocol [50]. Information regarding SB and PA modalities, as complementary information to the accelerometry, was measured using EMA. The use of EMA has been deemed suitable for use in a workplace context [56], and the questions employed were valid and reliable measure of SB and PA in adults [57]. Description of the EMA protocol is provided elsewhere [50].

#### 2.5.2. Work Engagement

The Utrecht Work Engagement Scale (UWES) (short-form UWES-9) administered using pen and paper, measured levels of work engagement using nine questions on a 7-point Likert-type scale (0–6) targeting the three constructs of vigour, dedication, and absorption [51].

#### 2.5.3. Anticipated/Perceived Intervention Benefits

Anticipated benefits in the domains of musculoskeletal and mental health, and work productivity prior to the intervention and after the intervention were measured using a questionnaire [50].

### 2.6. Qualitative Analyses

The focus group and interview data were analysed using a thematic approach, which allowed flexibility to systematically identify, organise, and offer insights into patterns of meaning, i.e., themes guided by the socio-ecological framework across the complete dataset in relation to the acceptability of the intervention [58]. Transcripts were read independently several times by two members of the research team (GN and CD) to undergo the process of familiarisation with the data, and to enable the creation of a set of preliminary codes. Line-by-line coding was then independently undertaken by GN and CD to assign the initial a priori themes and relevant excerpts. The codes were re-named according to the data collected. Initial codes were identified and applied to the data; any disagreements were discussed until consensus was reached. Inductive thematic analysis was also carried out which allowed for the emergence of additional themes. From the pre-defined and emergent themes, higher order themes were determined, forming a hierarchical structure. A process of moving back and forward between the entire dataset and the themes being produced, allowed iterative refining of the final higher order themes and subthemes. No software was used to code the qualitative themes. The Consolidated Criteria for Reporting Qualitative Research (COREQ) 32-item checklist was used in the reporting of the qualitative findings (Appendix A) [59].

### 2.7. Quantitative Analyses

As this was an exploratory feasibility pilot trial no formal sample size calculation was conducted [60]. The target sample size (*n* = 30) as determined by feasibility studies with similar aims [61,62], was decided upon based on pragmatic terms and the resource capacity available within the study. Descriptive analysis was used for recruitment, retention, and missing data. The Statistical Package for the Social Sciences V.25 (IBM Corp., Armonk, New York, NY, USA) and Microsoft Excel 2013 were used to analyse the quantitative data and to report descriptive statistics (mean, standard deviation, percentages). Inferential statistical tests were deemed not to be appropriate due to the exploratory nature of the feasibility trial [63]. Event-based outputs of SB and PA from activPAL files were entered into Excel spreadsheets with wake and work times, and sedentary, PA, and standing outcomes were extracted. Although some participants’ workday duration varied, the crossover design meant potential between-participant differences were controlled for as the same participants were involved in the intervention and control periods. The minimum data required for inclusion was four days of data, including at least one weekend day, for at least two of the three time periods. Inclusion criteria were set at this threshold in order to utilise the available data to analyse from small samples [64]. User-entered pedalling times on the Garmin Connect website were extracted and analysed. An acceleration threshold for the under-desk pedalling was developed using Microsoft Excel, i.e., cut-point threshold acceleration exceeding 375.0 Sum of Vector Magnitude, while seated (recorded as SB by activPAL3), and in bouts of ≥5 continuous minutes, was used to verify the pedalling times engaged in during working hours.

### 2.8. Progression Criteria to Full cRCT

Strict thresholds for progression criteria were not imposed, rather, a traffic light system with varying levels of acceptability was decided upon in the design phase, as recommended in studies that are exploratory in nature [65,66]. To inform progression criteria, Avery et al. [67] advocate that assessment of rates of completeness of outcome data is useful and important. Protocol adherence and completeness of outcome data were used as progression criteria. Validation of pedalling times in proportion to the goal of the intervention was used by the researcher to calculate protocol adherence rates.

Protocol adherence criteria:
Green—≥80% of participants achieved >60% of their pedalling goalAmber—60–79% of participants achieved >60% of their pedalling goalRed—<60% of participants achieved >60% of their pedalling goal.

Retention progression criteria:
Green—≥80% participants provided main trial-related outcomes (SB/PA) at T2Amber—60–79% of participants provided main trial-related outcomes at T2Red—<60% of participants provided main trial-related outcomes at T2.

## 3. Results

### 3.1. Feasibility

Of the 24 participants who expressed interest, three participants answered affirmatively to one/more of the rPARQ questions. Two participants refused to provide a letter from their doctor confirming their safety to participate. The recruitment target outlined in the protocol [50] (*n* = 30) was not achieved, with 73.3% of the target sample recruited. Full descriptive characteristics are shown in Table 1. In total, 81% of participants met the criteria for minimum wear time (4 days) providing accelerometery data in the intervention period, and 95% achieved minimum criteria for the control period. The 21 participants who completed the study provided 90% and 100% of data for work engagement, and musculoskeletal/mental health/work productivity intervention effects, respectively. For EMA measured data, 43% of surveys were completed. Twenty participants completed the focus groups. Of those who consented, 95% (*n* = 21/22) remained in the study until the end (Figure 2).

### 3.2. Acceptability

The initial analysis of all transcripts identified facilitators and barriers relating to the acceptability of the intervention from which emerged four higher-order themes (Table 2). As 20 of the 21 participants took part in the qualitative evaluation, data saturation was deemed to be achieved with 95% of the sample providing their user experiences and evaluation. These higher-order themes were identified as: intrapersonal (individual); interpersonal (social influences); environmental (prompts and pedal device) and organisational (work-related structures). Participants perceived and experienced the intervention in a predominantly positive manner, although some set-up issues were noted. The main individual level facilitators were education and awareness, sense of enjoyment, motivation and intention to potentially improve cardiovascular health, and a domino effect increasing PA outside of the intervention. Participants acknowledged that the education session had significantly increased awareness of the dangers of prolonged SB, whilst shock was expressed regarding the amount of PA required to attenuate the risks. A sense of enjoyment was perceived by many participants, both managers and employees, as a result of pedalling throughout the working day. Potentially improving heart health was expressed by one participant as a motivator to pedal while at work over the longer term. The men in the study described themselves as motivated and determined to achieve the goal of 30 min daily pedalling. Participants observed that the intervention ‘triggered’ new engagement in PA that had not previously been undertaken. This was particularly as a result of other data provided on the Garmin Connect website, such as steps, that were also being used as competition amongst some of the men. However, some participants expressed that at certain times of the working day they simply did not have the requisite time to engage in pedalling, and at particular times of the day work tasks were prioritised.

At an interpersonal level, many participants acknowledged that the social influence and *‘buddy vibe’* from co-participants was a powerful factor in promoting and motivating PA and reducing SB throughout the intervention period. The social group influence was important in terms of changing normative behaviours. It was felt by some participants that the shared activity of pedalling to reduce SB was beneficial to them, as it was sometimes perceived that other non-participating colleagues commented or displayed amusement at the men’s engagement in the intervention. Observational learning occurred when others engaged in the intervention activities. Many participants enjoyed the social comparison and competition element of the intervention, where the combination of the watch as well as the pedal machine fostered competitiveness in some participants, who continued the PA competitions into the evening times. Managers described a positive impact on the social environment and communicative aspects of work as a result of the intervention components. Those in management roles, in both office workplaces, reported that the intervention stimulated social interaction with colleagues, particularly *‘around the kettle’.* Some of this interaction was based around friendly rivalry.

Participants perceived the privacy of the under-desk cycle machine as a significant benefit to increasing PA in the professional workplace settings. This sentiment was expressed particularly by managers in both worksites. However, the enjoyment and ease of achieving the goal and pedalling throughout the day was very much predicated on the comfortable set up of the under-desk pedal machines. Correct set-up of the pedal device with the traditional desks was difficult to overcome for some of the participants, and taller participants found this more challenging to resolve. However, there remained a sense of appeal to using the pedal machines. Some participants experienced musculoskeletal relief from using the pedal machine to reduce their SB and increase PA in the workplace. On days of inclement weather or if other forms of exercise were not feasible, a major benefit was reported in having the pedal device available as an alternative. Overall, participants perceived the under-desk pedal machine to be useful due to the continuation of work tasks alongside pedalling throughout the day. The goal set in the intervention was deemed acceptable and possible to achieve. The ‘move’ bar on the Garmin watch was effective in reminding participants to break SB, both in the workplace, and at other sedentary times throughout the day. Participants, however, reported that the requirement to manually record the pedalling bouts on the Garmin watches was overly burdensome. In terms of productivity, the intervention was overall acceptable to management. The intervention did not adversely affect employees’ productivity, which was initial concern from an organisational perspective. This can be elucidated from the fact that managers discretely checked up on how employees were performing their work tasks while reducing their SB. It was also acknowledged that although productivity may have been affected when pedalling at high intensity speeds, slow intensity pedalling did not adversely impact on productivity. Overall, participants reported that they would continue with the intervention if assistance was provided in terms of the desk set-up, and pedalling bouts were automatically recorded in the mHealth component.

### 3.3. Acceptability of Measures—Ecological Momentary Assessment

Regarding the EMA, the repetitiveness of answering affirmatively to being sedentary resulted in some participants becoming less engaged and reactive to the EMA notifications, and participants described becoming a ‘*bit immune to it in the end actually.*’ The majority of participants perceived significant frustration as a result of ‘*constantly saying yes I’m sedentary*’. The nature of engagement in exercise meant that participants tended not to have their smartphone easily accessible and were not afforded the opportunity to record the various PA throughout their day, *‘when you’re actively doing exercise you actually miss it and time out so most of the time when you’re sedentary, not because most of the time you are sedentary but that’s when you actually see them* [the notifications].’ Some participants experienced disturbances as a result of the EMA notifications being too numerous in a busy workplace setting, *‘I get enough bloody notifications from all angles.’*

### 3.4. Acceptability, Appropriateness and Feasibility Questionnaire

Figure 3 illustrates the percentage agreement with questionnaire statements for participants (*n* = 21). In all but one statement across the three questionnaires, the median was 4 (IQR 0-1) indicating a consistent level of agreement with minimal dispersion in scoring between the participants. Statement 4 relating to the intervention’s feasibility, ‘the intervention seems possible’, scored the highest completely/agree of 86%. In all other statements, the level of equivalence (neither agree/disagree) was ≤33% and, in most cases (9/12 statements), the percentage prevalence was less than 25%. The highest level of disagreement was 24% in two statements, ‘the intervention seems implementable’, and ‘the intervention seems easy to use’. In all other statements (10/12 statements), the level of disagreement was ≤14%.

### 3.5. Sedentary Behaviour, Standing and Physical Activity

Table 3 presents the data in each period detailing the outputs from the activPAL3 data. Results showed a decrease in workday SB from 379.3 (SD 79.0) to 358.9 (SD 96.6) minutes per working day in the intervention period compared to the control period; thus, an indicative reduction of workplace sedentary behaviour of 20.4 min-per-workday. Total weekday SB reduced from 634.5 (SD 102.5) to 588.8 (SD 107.8) minutes per day in the intervention group compared to the control group, indicating a 45.7-min reduction in sedentary behaviour. In terms of physical activity (i.e., stepping), average total weekday PA increased by 9.9 min in the intervention period compared to the control period. Overall, workday standing increased by 14.4 min per day in the intervention period, while total weekday standing increased by 23.2 min per day.

### 3.6. Work Engagement

Work productivity data are presented in Table 4. Minimal differences were observed over the duration of the study for work engagement.

### 3.7. Anticipated and Perceived Intervention Benefits of the Intervention

Mean scores (*n* = 21) for anticipated improvements to back/neck pain, mental health, and work productivity because of reducing SB in the intervention all resulted in an average of 4 (agree) at baseline (scored 1–5, 1 being strongly disagree, 5 being strongly agree). The mean score for the perceived intervention benefits to mental health remained at 4.0 (agree) at follow-up, indicating that participants agreed that the intervention *would* benefit mental health, and further agreed that it *did* benefit mental health at follow-up. Perceived benefits to work productivity and back/neck pain at follow up were 3.0, which represents ‘neutral’ on perceptions of improvements of work productivity and back/neck pain after the intervention.

### 3.8. Pedaling Activity and Adherence to the Protocol

Figure 4 illustrates daily pedalling times in minutes for each day of the intervention. Participants pedalled an average 27.1 ± 10.23 min-per-workday in the intervention period. Overall, 67% of participants engaged in >20 min of pedalling per day, which equated to >60% of the intervention pedalling goal.

### 3.9. Progression Criteria

Goyder and colleagues [68] have advised that reporting data completeness is an integral part of clinical trial and intervention reporting. Hence, the summary of data completeness is shown on a CONSORT flow chart from participants’ enrolment, and at all time points in the study. The completeness of the main-trial related outcome data collected was high in the intervention period (90%), and in the control (81%). Overall, this would indicate a ‘green’ situation as per the stated progression criteria. In assessing the reasons for missing data using the activPAL3, it was determined that the issues could be resolved in a future trial.

## 4. Discussion

The aim of this study was to examine the acceptability and feasibility of a multicomponent intervention to reduce SB and increase LPA in professional men. Secondary aims were to provide preliminary changes in SB, standing, PA (including pedalling times), work engagement, and benefits to musculoskeletal and mental health, and work productivity following the intervention. The trial was feasible to deliver in this cohort, with a very low dropout rate, and successful collection of outcome variable data. High missing EMA data were reported, and this method was found to be overly burdensome for participants. The qualitative findings suggested overall acceptability of the intervention, however future iterations could be improved in two areas which centred on the ergonomic set-up of the pedal machine and the burden of manually recording pedalling times. Nonetheless, preliminary data indicate that this multicomponent intervention may improve SB by replacing it with LPA accrued during the working day, with minimal impact on work engagement and productivity.

The trial was feasible to implement in professional male employees, including managers and managing partners. Compliance with providing outcome measures was high, and the trial-related measures and study processes were overall acceptable to this target group as found in the qualitative component of the evaluation, in which 95% of participants completed. Regarding protocol adherence, just over two thirds of participants achieved more than 60% of their average daily pedalling goal in the intervention period, although daily pedalling times reduced as the intervention progressed. This finding is similar to adherence reported by Peterman et al. [69]. The target sample outlined in the protocol was not achieved, however, it scored favourably compared with other feasibility studies with similar aims of predominantly women participants (57%) [61]. Retention in the study was very good, also higher than reported in similar studies (86%) [61]. This strengthens the assertion that initial contact with managers may be a useful facilitator in recruitment to workplace interventions. The gender-sensitive approach in this study may have resulted in increased recruitment and retention rates. Previous studies have reported that mixed-gender health promotion initiatives have sometimes failed to engage men [41]. The findings of this study extend previous evidence where gender-sensitive programmes have been found to be somewhat feasible in rural workplaces (retention rate 58%) [43], by highlighting the potential of a men-only intervention in an urban location.

Qualitative insights from participants suggested that increased awareness of the dangers associated with prolonged SB provided by the researcher via the education session at the outset of the study was a major facilitator to intervention engagement. This finding has been reported in previous studies concerned with reducing SB [61,70,71]. Educating participants can increase consciousness of their own SB and has been found to create shock about potential health consequences of prolonged SB. This is consistent with a lack of knowledge of health risks associated with SB reported in the general population [70]. Awareness was further heightened by the weekly feedback on participants’ personalised patterns of SB. In line with Brakenridge et al. [71], this minimally intensive approach was rated as valuable to participants. The current study extends the literature by demonstrating the facilitating motivator of education and feedback in reducing SB in professional males. Furthermore, the findings strengthen previous research highlighting the value of adopting a gender-sensitive approach to engage and retain men in health promotion interventions [72,73]. The evidence provided by this study adds to the literature by investigating previously untested intrapersonal gender-sensitive techniques in a workplace intervention to reduce SB, in particular, knowledge and awareness, and goal-setting. Some participants were keen to engage in LPA using the pedal machine to improve cardiovascular health by elevating their heart rate in the longer term. Public health and health promotion campaigns seek to improve men’s heart health, in particular the prevention of cardiovascular disease [46]. Accumulating PA and increasing heart rate in a workplace setting could be used as a target to reduce the disease burden in men.

The strength of the influence of the social environment reflected in the findings of this study is supported by the literature [74,75,76,77]. Social relations are of upmost importance in influencing workplace behaviour. Targeting social interaction has improved compliance in workplace interventions that focus on PA [78], and now with SB as observed in the present study. The intervention components and study topic provided new opportunities for social interactions. In a study using sit-stand desks, Dutta et al. [79] reported an increase in ‘social energy’ and enjoyment of face-to-face interactions from employees. Participants in the present study enjoyed this topic of conversation, and managers in particular expressed this sentiment. An explanation may be that hierarchical structures may inhibit more relaxed conversation, and the intervention may have provided an opportunity to discuss a shared topic common across work roles. Changing what is considered normal workplace behaviour is likely to be a key facilitator of large-scale behavioural change to reduce occupational SB and increase LPA. Cultural change, not only in terms of individual behaviour, but equally environments to facilitate change, policies, and leadership are required. This important finding strengthens the literature by highlighting that recruitment of managers to participate in interventions may be a valuable and supportive strategy to study engagement.

Social comparison targeted the mechanism of action, social influence [80]. Participants reported that observation of peers’ engagement in the intervention was a strong ‘*catalyst*’ to increase PA. The importance of collegiality was expressed which indicates that this may have been more than a simple prompt. Peer pressure and social support encouraged workplace PA. It appears that observational learning is important to reduce SB, as employees learn and conform to the behaviour of the majority and are concerned about how behaviour outside of this norm is perceived. O’Dolan et al. [61] found that observational learning was an important construct to reduce occupational SB in bank employees, and together with the findings of the present study, highlights that this is an important target for future interventions to reduce occupational SB. Friendly competition was a key part of the intervention design. Websites and mobile apps provide a valuable medium for social support and friendly competition within workplace team-based programmes, where teams of male peers can compare their progress using virtual platforms [41,81]. Competition also improves compliance with wearing of activity trackers [82]. Minimal removals of the tracker watch were reported in the present study, further strengthening this evidence. Data collected by the Garmin watch, such as the accruement of daily steps was also used by some employees as scope for competition. Although some participants chose not to engage with the Connect platform, many found the strategies of self-monitoring and social comparison via the competition element to be a ‘*driver of incremental activity*’, not only at work but throughout the day and evening times. This strengthens the literature that social comparison and competition strategies are useful motivators of PA in professional males [41]. Furthermore, the findings extend the literature on the use of mHealth to target these strategies by providing important information on the acceptability and feasibility of harnessing mHealth in workplace interventions to reduce SB in professional settings [82]. A sense of togetherness and social support of others and the ‘*buddy vibe*’ fostered by engagement in the intervention reduced self-consciousness of pedalling in the professional workplaces. In terms of normative behaviours, a concern outlined in the development phase was a fear of being perceived as ‘*weird*’. Some pressure from non-participants in terms of deviance from social norms was reported. This type of social judgement and the importance of colleagues’ perception of oneself as being ‘normal’ has been reported in previous studies [83]. It appears that within each worksite the requisite number of people were involved in the study to enable a group effect. This is an important finding in intervention development research, and supports the adoption of the socio-ecological model to target important factors of influence, in particular the social context [6], that was particularly relevant to the professional males in this study.

Participants enjoyed pedalling the under-desk device, however, the foundation of this enjoyment was based on comfortable desk set-up within the participants’ office workstations. Significant difficulties were reported by some in attaining positions to allow pedalling behaviours without knocking knees on the desks, which is reflected in previous studies [84,85]. To establish that the floor to desktop height had the adjustability required to use the Deskcycle^TM^, dimensions were obtained from the worksites in the development phase, however, even following adjustments to increase the desk height, some taller participants still found positioning issues too difficult to overcome. Despite the barriers to pedalling, participants endeavoured to engage in the intervention activities and to achieve their pedalling goals, demonstrating perseverance to reduce occupational SB and strong engagement with the study. The utilisation of the pedal device as an alternative strategy has been reported previously [34], and highlights the potential benefit and convenience of this type of device in a workplace context as it enables new ways of increasing occupational PA. Future studies employing this device could aid participants to ensure comfortable ergonomic positioning. Using height-adjustable desks may allow more leg clearance room for pedalling [86]. The findings add to previous studies which investigated the acceptability of this type of device with women [84] by highlighting issues of acceptability and feasibility in male participants. The use of prompts to encourage breaks in sitting has produced promising results in intervention studies [14,15,16,17]. Digital prompts have been found to be more effective than education alone at reducing occupational SB [87]. This finding was strengthened by evidence that the move bar on the tracker watch in the present study acted as an external digital prompt to increase movement and break SB. The restructuring of the environment in the direct and proximal area of each participant’s workstation with the intervention equipment also served to act as a visual and physical reminder to participants to engage in PA.

Recruitment of managers to the study, similar to Healy et al. [88], demonstrated that organisational support for reducing SB is essential in successful multicomponent interventions. Culture at the organisational level includes values, norms, structures, operations, strategy and policy that operate in a dynamic and non-static way to impact employees’ opportunities and tendencies towards moving more at work [40]. Embedded in the facets of organisational culture are explicit and implicit orientations towards physical inactivity and SB. Although an organisation may explicitly declare goals to improve employees’ wellbeing, when the opportunities to reduce SB centre on moving away from the workstation, an implicit pressure may be felt by employees surrounding a perceived reduction of productivity [78]. Recruiting managers to participate in the intervention was an effective strategy to promote a supportive culture at an organisational level, and endorsed wellbeing values through modelling behaviours of senior management, thus adding to this body of literature [71,89]. From an employee perspective, participants acknowledged a significant advantage in the combination of pedalling while conducting work tasks. Reading documents and speaking on the telephone were particularly suited to pedalling. Work performance was affected as pedalling intensity increased, which resulted in some productivity issues. In congruence, Tronarp et al. [90] reported that light intensity pedalling only slightly impaired work performance, compared to moderate intensity pedalling which affected work performance more significantly. Importantly, from a management perspective the intervention did not negatively impact on productivity levels. These findings demonstrate that low intensity physical activity can be conducted in a workplace context whilst not reducing work performance capacity.

At baseline, participants spent on average 10.33 ± 1.76 (mean ± SD) hours per day of their waking hours engaged in sedentary behaviour. During working hours, the average duration of SB the men engaged in was 6.72 ± 1.85 h per working day. These findings strengthen the literature demonstrating that males [91], in desk-based or white collar employees engage in dangerous levels of SB [91,92,93]. Although the present study was not powered to conduct inferential statistics, indicative reductions of SB of −45.7 min per total weekday and −20.4 min per workday were found in the intervention compared to the control period. This is similar to previous findings of a multicomponent intervention using a pedal machine with access to a motivational website, predominantly women (90%), where 58.7 min reduction of daily SB with using was reported [84]. Similarly the 27.1 min pedalling time per day reported in the current study is in line with the 31.1 min/day reported by Carr et al. [84]. Importantly, the present intervention enabled an increase in workplace activity, without a compensatory decrease in PA for the remainder of the day. A decline in pedalling times in the second week of the intervention was observed, however, a significant burden was reported with the manual recording of pedalling bouts using the Garmin watch. Although this method was used a self-monitoring technique, it was overly burdensome in a busy professional workplace. Future studies should provide automatic technology to record bouts of pedalling to allow participants to start and stop whenever it is suitable for them in their working day.

The mental health effect of the intervention was viewed favourably by the men, with agreement that the intervention *would* benefit mental health, and further confirmation that it *did* benefit mental health at follow-up. These results strengthen the positive findings within mixed evidence reported in a recent literature review investigating the effectiveness of workplace interventions on well-being [94]. The findings also add to the literature suggesting the positive mental health impact of digital workplace interventions [95].

The positive response to anticipated improvement of back/neck pain as a result of the intervention decreased to a ‘neutral’ score at post-intervention follow-up. Although the scores decreased from pre- to post intervention time points, it may be argued that although participating in the intervention did not improve neck/back pain, it importantly did not induce back/neck pain. Mixed findings have been reported regarding the association of musculoskeletal issues and prolonged stationary sitting [96,97]. Similarly, conflicting results have been found in terms of what impact, if any, intervention strategies used to reduce SB have on musculoskeletal symptoms, in terms of participant comfort, or the health benefits associated with each strategy [98,99,100,101].

Similarly, the score for productivity in the present study decreased from pre- to post intervention, however, the ‘neutral’ response demonstrates that although the intervention did not improve productivity, neither did it reduce work productivity. This may be important as workplace pedalling, compared with treadmill and standing workstations, allows employees to experience greater cardiometabolic gains, together with the maintenance of acceptable levels of productivity in work performance [102]. Increasing PA throughout the working day can contribute to increased productivity and reduction in injuries and absenteeism [103], which may be particularly advantageous in a corporate environment. The results of this study add to the literature in highlighting the importance of the physical and mental benefits, as well as work productivity of the provision of a pedal machine to professional men.

### Strengths and Limitations

The main strength of this study was its mixed methods approach to evaluation. Another major strength was application of the socio-ecological model and the behaviour change strategies used in the development and evaluation of the intervention. The waitlist crossover design was a strength as participants acted as their own control and thereby reduced between group differences. Objective measures of SB and PA were collected using a device-based instrument. By exploring the acceptability of a multicomponent intervention with professional men, in varying roles e.g., employees, managers, and managing partners, practical improvements to the intervention were ascertained, which may be incorporated to inform the development of a fully powered cluster RCT.

The results of the pilot study should be interpreted in the context of several limitations. The sample size target outlined in the protocol was not achieved. It is also unlikely that the wash-out period negated the effects of the education regarding the dangers of SB, and thereby possibly affected behaviours in the control period. It may be argued that those who consented to participate were more motivated to reduce their SB than the general population, indicating selection bias. Blinding of the participants and researchers was not possible due to the nature of the trial, however, the use of objective outcome measures minimised researcher bias.

## 5. Conclusions

For many adults, the workplace is a key setting to increase PA and reduce SB. Results of this study suggest that it was somewhat acceptable and feasible to implement a theory-led multicomponent intervention to reduce SB and increase LPA in a workplace setting with professional men. The Cycle at Work intervention has the potential to elicit change in SB by increasing LPA, however, due to the small sample size, results should be treated with caution, and a RCT with a larger sample size, and including women, is required to confirm these findings. Careful consideration of the ergonomic set up and automatic technology to record the pedalling bouts needs to be incorporated in future trials of this intervention before being used on a larger scale. Development and dissemination of national guidance, together with the promotion and implementation of workplace health programmes, are required to increase PA, reduce SB, and promote incidental PA during the working day for employees. These workplace programmes need be implemented in different occupations and settings, and with a priority focus on the least active. The findings enhance the knowledge base, and highlight the opportunities and challenges met in the process of conducting this intervention which may be of benefit to future investigators of workplace interventions to reduce SB.

## Figures and Tables

**Figure 1 ijerph-18-09292-f001:**
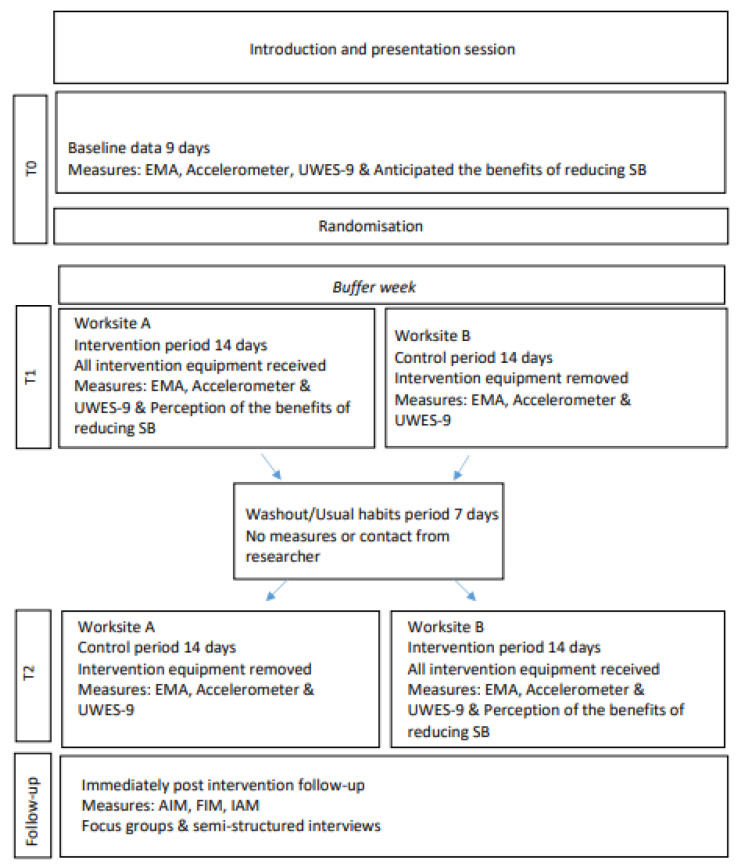
Study and participant flow diagram.

**Figure 2 ijerph-18-09292-f002:**
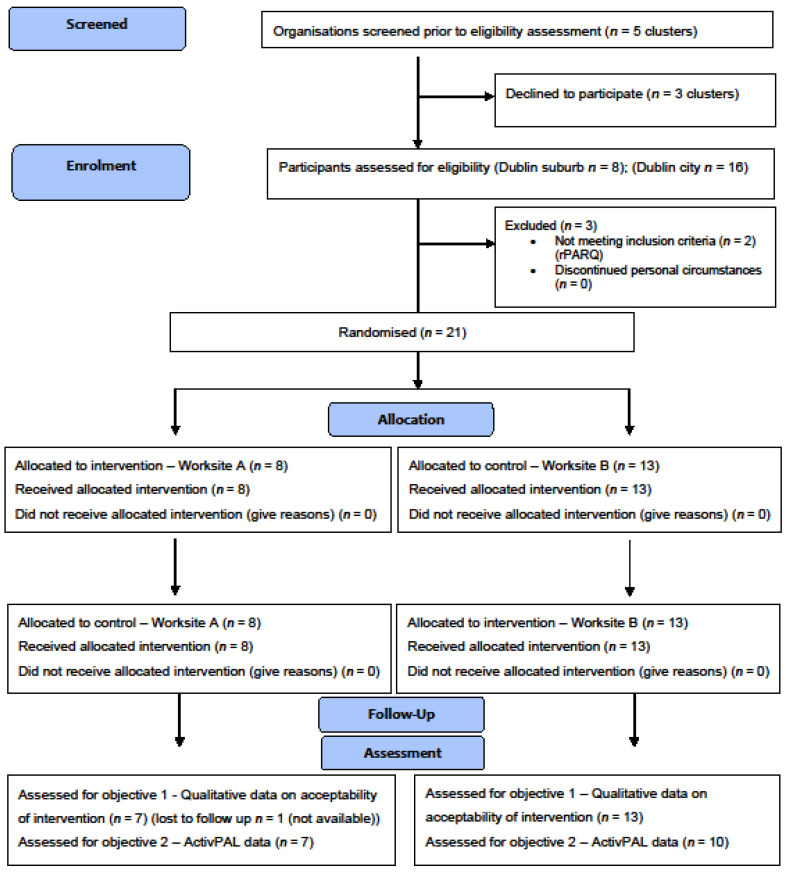
Adapted CONSORT flow diagram illustrating participant retention [54].

**Figure 3 ijerph-18-09292-f003:**
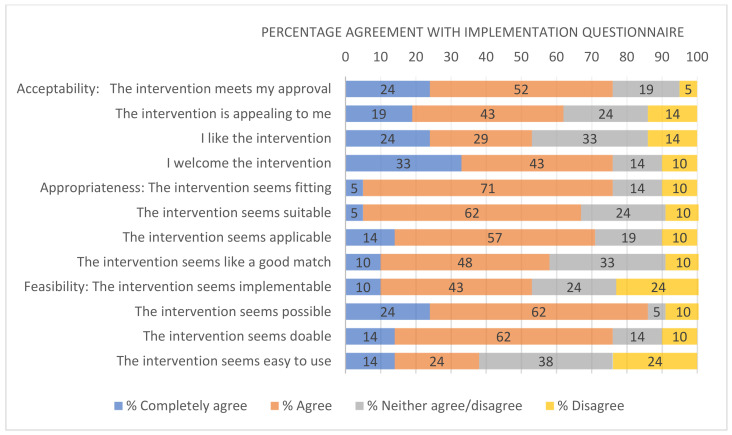
Percentage agreement with implementation questionnaire. There were no responses in the completely disagree category.

**Figure 4 ijerph-18-09292-f004:**
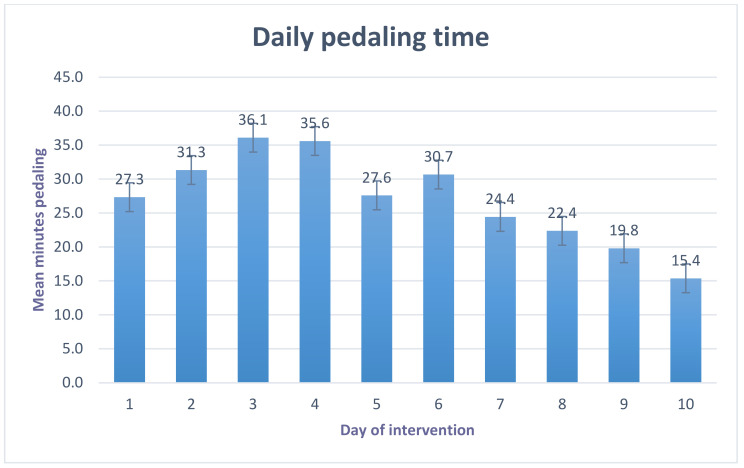
Minutes of pedalling time per day of the intervention.

**Table 1 ijerph-18-09292-t001:** Participant characteristics.

Characteristic	Worksite A	Worksite B	Total
Type of company	Online training	Legal firm	
Location	Dublin suburb	Dublin city centre	
Total participants	8	14	22
Total no. managers	1	7	8
Total no. of employees	7	7	14
Mean age years (SD)	44.4 (11.0)	41.5 (11.0)	42.9 (11.0)
Hours worked per week (SD)	41.1 (4.1)	44.0 (10.5)	42.6 (7.3)
Workday sedentary behaviour min (*n* = 20)	399.7 (36.8)	406.7(141.1)	403.6 (111.2)
Total weekday sedentary behaviour	630.7 (82.4)	611.3 (115.4)	619.9 (105.5)
Total weekend sedentary behaviour	560.3 (85.0)	467.1 (81.7)	508.6 (97.9)
Workday physical activity	37.6 (7.8)	50.1 (12.7)	45.1 (12.9)
Total weekday physical activity	79.9 (18.6)	102.9 (21.3)	93.7 (23.8)
Total weekend physical activity	122.2 (64.7)	136.0 (38.3)	130.5 (52.3)
Workday standing	73.4 (16.4)	122.5 (89.7)	102.9 (76.2)
Weekday total standing	171.6 (31.0)	225.9 (115.02)	204.2 (97.5)
Weekend standing	220.8 (64.7)	241.8 (57.3)	233.4 (62.8)

**Table 2 ijerph-18-09292-t002:** Facilitators and barriers of Cycle at Work intervention.

Intervention Facilitator Themes	Quotes
*Individual*	
(a) Knowledge, education, and awareness	(a) ‘I do agree with the commentary and the awareness feature or the factor of raising awareness because it’s on your wrist, it’s under your desk, it’s on your screens, and it’s on your phone so it did make me very mindful of the need for activity.’ [P4, Manager]
(b) Sense of enjoyment	(b) ‘Actually I found the days I did it I found it quite a nice thing to have done.’ [P4, Manager]
(c) Motivation to improve cardiovascular health	(c) ‘I liked the idea of raising my heart rate while I was working and if we can get the set up right I’d be very interested in doing that long term.’ [P1, Employee]
(d) Domino effect	(d) ‘I started noticing fellas out that I never saw walking before. Because what it was doing was it was triggering other practices where they knew that they were on a timer, you know Liam being an example every day.’ [P2, Manager]
*Interpersonal*	
(a) Sense of togetherness	(a) ‘Is the catalyst for that the fact that your peers are all doing it or is it that you are self-conscious that you know I am sitting too much during the working day? Because I think it is more the former than the latter.’ [P5, Manager]
(b) Observational learning	(b) ‘Yeah but it was just remembering to do it I suppose was the main issue you know. Like if someone else in the office I heard kicking it off then I would go oh yeah they’re doing that and that would trigger it.’ [P3, Employee]
(c) Social comparison	(c) ‘It was good, certainly I noticed more competitiveness with different people, they were certainly way more competitive than I thought they should have been, to extremes, I think but not in a bad way but it was interesting watching it unfold.’ [P2, Manager]
(d) Opportunity for social interaction	(d) ‘I mean from a management perspective I suppose to the extent that it does engender a sense of competitiveness whether they see it on the app or they start talking about it which was great and actually the fact that we’re a cross section in the office you know we had a whole different things to talk to and grill the lads about you know.’ [P5, Manager]
*Environmental*	
(a) Privacy of under-desk pedalling	(a) ‘Yeah because like that privacy and semi privacy thing can say well you know you’re more likely to use the machine.’ [P1, Manager]
(b) Musculoskeletal improvement	(b) ‘One of the questions on the survey is did you get any improvement in back and neck pain and I actually never thought of that until that question came up and then I thought yeah, it has improved a little bit.’ [P2, Manager]
(c) Use of pedal device as alternative	(c) ‘I was staying in at lunchtime having that there helped to be able to chip away on or if the weather was miserable outside and it wasn’t great to go out for a walk that was definitely beneficial.’ [P1, Employee]
(d) Complementation of work tasks while pedalling	(d) ‘Yeah but like 40 min, like everyone’s job is different but you know it’s not a huge amount of time over the course of a day…you know you can do your typing, you can do a call, you know you can do your reading, it’s not that you’re going so fast that you can’t do your tasks.’ [P1, Manager]
(e) Move bar prompted movement	(e) ‘One thing I did find very good at home but obviously also in work was if you’re not moving for a certain amount of time it (Garmin ‘move bar’) sends you the little arrow to move which was good.’ [P3, Employee]
*Organisational*	
No detrimental effect on productivity	‘But I was more concerned about what are the lads doing inside in the room, when are they doing it and is it disrupting their productivity, so I have to say every time I kind of went in I wasn’t like, I was just going into the room, but they were doing it as they were working.’ [P2, Manager]
**Intervention barriers**	
*Individual*	
Time priorities	‘Like some mornings I found, I don’t know if you did it too, I actually just kicked it out of the way just because the first couple of hours I just didn’t want to be dealing with it. But otherwise it was grand.’ [P3, Employee]
*Interpersonal*	
Social judgement	‘I think it was good that there was a bunch of people doing it because you can see others using it and you get your steps in and people would tend to walk past the office and just laugh.’ [P1, Employee]
*Environmental*	
(a) Ergonomic set-up	(a) ‘If there was a way to make it a little bit more user friendly to someone like me or the facilities that we have I’d have no issue doing it. I actually love the concept of it, I just think that there’s a few tweaks that need to be done to make it sort of more appealing.’ [P1, Manager]
(b) Garmin watch manual- recording	(b) ‘I think if it wasn’t timed, I would have done a little bit more because you have to remember to actually time it and there was a little bit of setting it up.’ [P1, Employee]

**Table 3 ijerph-18-09292-t003:** Means and SDs in secondary outcomes between intervention and control periods.

	Intervention (I) *n* = 17	Control (C) *n* = 19	Difference C-I
Workday SB minutes (SD)	358.9 (96.6)	379.3 (79.0)	−20.4
Total weekday SB	588.8 (107.8)	634.5 (102.5)	−45.7
Weekend sedentary behaviour	498.9 (108.4)	507.7 (106.4)	−8.9
Workday physical activity	48.7 (13.8)	48.5 (13.8)	0.3
Total weekday PA	103.2 (29.2)	93.4 (24.3)	9.9
Weekend physical activity	124.0 (38.7)	125.5 (36.1)	−1.5
Workday standing	110.1 (72.1)	95.7 (36.2)	14.4
Total weekday standing	219.7 (94.7)	196.5 (52.0)	23.2
Weekend standing	239.4 (62.5)	229.3 (58.7)	10.1

**Table 4 ijerph-18-09292-t004:** Means and SDs in secondary outcomes between intervention and control periods.

		Intervention		Control
	*n* = 21	Mean (SD)	*n* = 19	Mean (SD)
Work engagement (total)		4.23 (0.8)		4.33 (0.8)
Vigour		3.94 (1.1)		4.16 (0.9)
Dedication		4.44 (0.8)		4.49 (0.8)
Absorption		4.32 (0.8)		4.35 (0.9)

## Data Availability

The data used for this study is available upon request.

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
