# Peer review of "A Cluster-Randomised Crossover Pilot Feasibility Study of a Multicomponent Intervention to Reduce Occupational Sedentary Behaviour in Professional Male Employees"

_ijerph, 2021, doi:10.3390/ijerph18179292_

Round 1
Reviewer 1 Report
Title:
Suitable
Abstract
Line 13: the method used in the research should be specified in more detail.
Line 14 : very small sample. It is advisable to increase the sample. In addition, it would be interesting to include women in the study.
Line 14 : the mean age of the sample must be specified.
Introduction
Lines 49 y 50: this statement should be treated with caution, as this connotation may vary depending on the type of feedback we are referring to in the manuscript (interrogative, descriptive, etc.). The type of feedback should perhaps be specified, or, failing that, some example of it should be included.
Lines 81 to 85: If we continue to pay attention to these statements, we will not be helping to modify the habits traditionally established in both male and female adults. It is true that there are generally these differences in behavior between both genders, but we should not take certain thoughts for granted, but on the contrary, once known, to address them from their root to design programs that, taking this into account, can modify the habits acquired by others that improve health and therefore the quality of life of all people.
Materials and Methods
Participants
The study sample is not specified
Study design and procedures
Suitable
Results
Line 308: the sample is very small, it is recommended to increase it. In addition, it would be interesting to include women in the study.
Line 410: the foot of the figure should start on the left, not centered.
Discussion
Suitable
Conclusions
It should be reported that, due to the small sample size of the study, the results should be treated with caution.
Author Response
The authors would like to thank the reviewer for their careful reading and consideration of the manuscript, and their subsequent comments and suggestions which have substantially improved the manuscript.
Reviewer 1
Title:
Suitable
Abstract
Point 1: Line 13: the method used in the research should be specified in more detail.
Response 1: Line 13-15, pg.1: The methods used in the study have now been specified in more detail.
“The main elements of the intervention comprised: a Garmin watch with associated web-based platform/smartphone application; an under-desk pedal machine; and management participation and support. A cluster-randomised crossover pilot feasibility trial recruiting professional males was conducted in two workplaces. Mixed methods were used to assess the primary outcomes of recruitment, retention, and acceptability and feasibility of the intervention.”
Point 2: Line 14 : very small sample. It is advisable to increase the sample. In addition, it would be interesting to include women in the study.
Response 2: Lines 16-18, pg. 1: This was a small pilot study with primary outcomes of assessing acceptability and feasibility of the intervention. It would be hoped in the future to conduct a definitive RCT using a larger sample size and which would also include women.
Point 3: Line 14 : the mean age of the sample must be specified.
Response 3: Line 20, pg.1: The mean age of the participants has now been included in the abstract.
“Twenty-two participants were recruited (mean age 42.9 years (SD 11.0).”
Introduction
Point 4: Lines 49 y 50: this statement should be treated with caution, as this connotation may vary depending on the type of feedback we are referring to in the manuscript (interrogative, descriptive, etc.). The type of feedback should perhaps be specified, or, failing that, some example of it should be included.
Response 4: Line 52-56, pg.2: Different types of feedback is now specified in this section, as well as an example of descriptive and persuasive feedback.
“Irrespective of the target behaviour or technology used in intervention studies, different types of feedback have been found to be an effective way to change habits [10]. For example, van Dantzig et al. [13] provided descriptive and persuasive feedback to participants’ smartphones whenever 30 minutes of uninterrupted computer activity (used as a proxy for SB) was recorded, to break their SB.”
Point 5: Lines 81 to 85: If we continue to pay attention to these statements, we will not be helping to modify the habits traditionally established in both male and female adults. It is true that there are generally these differences in behavior between both genders, but we should not take certain thoughts for granted, but on the contrary, once known, to address them from their root to design programs that, taking this into account, can modify the habits acquired by others that improve health and therefore the quality of life of all people.
Response 5: Lines 91-93, pg.3: We thank the reviewer for making this interesting and important point, and it is agreed that the root causes of occupational sedentary behaviour are similar for both genders. This has now been included as follows:
“Although the root causes of occupational SB are similar for both genders - i.e., restrictive workstations and the traditional workplace culture of remaining seated for long periods - in terms of intervention participation, men are especially difficult to recruit to health promotion interventions [41].”
Materials and Methods
Participants
Point 6: The study sample is not specified
Response 6: The sample size is now specified in the Participants section.
Line 130-131, pg.3: “Twenty-two office-based employees from two professional worksites in Dublin, Ireland were recruited.”
Study design and procedures
Suitable
Results
Point 7: Line 308: the sample is very small, it is recommended to increase it. In addition, it would be interesting to include women in the study.
Response 7: it is hoped to conduct a larger study to test the effectiveness of the intervention to reduce sedentary behaviour and increase physical activity in both males and females, and in a range of workplace settings. This is now outlined in the Conclusions section [Lines 701-706, pg. 20-21].
“The Cycle at Work intervention has the potential to elicit change in SB by increasing LPA, however, however, due to the small sample size, results should be treated with caution and a RCT with a larger sample size, and including women, is required to confirm these findings.”
Point 8: Line 410: the foot of the figure should start on the left, not centered.
Response 8: Line 414-415: The footer of Figure 3 has been moved to the left.
Discussion
Suitable
Conclusions
Point 9: It should be reported that, due to the small sample size of the study, the results should be treated with caution.
Response 9: Line 701-706, pg.20-21: This addendum has now been included in the conclusions section.
“The Cycle at Work intervention has the potential to elicit change in SB by increasing LPA, however, however, due to the small sample size, results should be treated with caution and a RCT with a larger sample size, and including women, is required to confirm these findings.”
Reviewer 2 Report
This paper is insightful and scientifically sound. Please see and consider my comments below.
Line 56: An emphasis on the frequency as well as the prescribed time of breaks could be a good addition to this sentence.
Line 90: Please remove the bracket after from
Besides the improvement of the physical and mental health, it could be interesting to further elaborate on the advantages that the promotion of breaks within a corporate environment could bring to the productivity and the retention of employees.
Line 330: Was the cardio-vascular health evaluated and monitored?
Table 2 could be re-formatted in a more informative and intuitive way.
Figure 4: Error bars should be added to the histogram
A comparison of the outcome of this study to similar ones could be a good addition to the manuscript.
Author Response
Reviewer 2
This paper is insightful and scientifically sound. Please see and consider my comments below.
The authors would like to thank the reviewer for their careful reading and consideration of the manuscript, and the subsequent comments and suggestions which have substantially improved this paper.
Point 1: Line 56: An emphasis on the frequency as well as the prescribed time of breaks could be a good addition to this sentence.
Response 1: Lines 58-64, pg.2: An emphasis on interruptions to and duration of breaks to prolonged SB, as well as the metabolic effects as outlined in the literature has now been included in this section.
“Prolonged SB triggers a state of metabolic ‘inflexibility’, even among individuals who meet PA recommendations, by disrupting fuel homeostasis and metabolic health. Frequent interruptions to SB with bouts of activity (even 1 minute duration) have been associated with improved metabolic outcomes, including in those who exercise regularly [20,21]. Thus, breaking up time in SB is a stimulus for improving metabolic health (flexibility) and has been suggested as a novel and promising strategy in the general population [22].”
Point 2: Line 90: Please remove the bracket after from
Response 2: The bracket has now been removed from this sentence.
Point 3: Besides the improvement of the physical and mental health, it could be interesting to further elaborate on the advantages that the promotion of breaks within a corporate environment could bring to the productivity and the retention of employees.
Response 3: Lines 673-676: The beneficial effects of increasing physical activity and decreasing sedentary behaviour in terms of increased productivity and reduced absenteeism has now been added to the Discussion section.
“Increasing PA throughout the working day can contribute to increased productivity and reduction in injuries and absenteeism [107], which may be particularly advantageous in a corporate environment. “
Point 4: Line 330: Was the cardio-vascular health evaluated and monitored?
Response 4: Cardiovascular health was not evaluated and monitored in this pilot study.
Point 5: Table 2 could be re-formatted in a more informative and intuitive way.
Response 5: Table 2 has been re-formatted to separate the quotes in a more delineated manner which has resulted in more informative and intuitive table.
Point 6: Figure 4: Error bars should be added to the histogram
Response 6: Figure 4: Error bars have now been added to the histogram.
Point 7: A comparison of the outcome of this study to similar ones could be a good addition to the manuscript.
Response 7: We would like to draw the reviewer’s attention to each of the elements of acceptability and feasibility- the primary outcomes - in the Discussion section and the comparisons to similar studies made.
Lines 498-499, pg.17: Comparison to a similar study in terms of adherence to the protocol has now been added.
“Regarding protocol adherence, just over two thirds of participants achieved more than 60% of their average daily cycling goal in the intervention period, although daily cycling times reduced as the intervention progressed. This finding is similar to adherence reported by Peterman et al. (69).”
Recruitment and retention rates in the current study are compared to similar studies.
Lines 499-504, pg.17: “The target sample outlined in the protocol was not achieved, however, it scored favorably compared with other feasibility studies with similar aims of predominantly women participants (57%) [61]. Retention in the study was very good, also higher than reported in similar studies (86%) [61]. This strengthens the assertion that initial contact with managers may be a useful facilitator in recruitment to workplace interventions.”
Acceptability of the various elements of the intervention, and of the study overall is presented throughout the Discussion section.